# From fake to real: Pretraining on Balanced synthetic images to prevent Bias

## Abstract

Visual recognition models are prone to learning spurious correlations induced by a biased training set where certain conditions $B$ (*e.g.*, Indoors) are over-represented in certain classes $Y$ (*e.g.*, Big Dogs). Synthetic data from generative models offers a promising direction to mitigate this issue by augmenting underrepresented conditions in the real dataset. However, this introduces another potential source of bias from generative model artifacts in the synthetic data. Indeed, as we will show, prior work uses synthetic data to resolve the model's bias toward $B$, but it doesn't correct the models' bias toward the pair $(B, G)$ where $G$ denotes whether the sample is real or synthetic. Thus, the model could simply learn signals based on the pair $(B, G)$ (*e.g.*, Synthetic Indoors) to make predictions about $Y$ (*e.g.*, Big Dogs). To address this issue, we propose a two-step training pipeline that we call From Fake to Real (FFR). The first step of FFR pre-trains a model on balanced synthetic data to learn robust representations across subgroups. In the second step, FFR fine-tunes the model on real data using ERM or common loss-based bias mitigation methods. By training on real and synthetic data separately, FFR avoids the issue of bias toward signals from the pair $(B, G)$. In other words, synthetic data in the first step provides effective unbiased representations that boosts performance in the second step. Indeed, our analysis of high bias setting (99.9%) shows that FFR improves performance over the state-of-the-art by 7-14% over three datasets (CelebA, UTK-Face, and SpuCO Animals).

## 1 Introduction

Visual recognition models are prone to learning spurious correlations (Bias) (Wang et al., 2020a; Zhao et al., 2021; Meister et al., 2022). These correlations frequently arise due to an imbalance in the training set. For example, given a dataset with classes $Y$ (*e.g.* Smiling vs Not Smiling), there exists a confounding bias variable $B$ (Gender: Male and Female) in the training set such that one bias group (*e.g.* Male) is represented in one class more than others (*e.g.* most males are Smiling). This leads models to use the bias signal $B$ (gender) mistakenly to predict $Y$ (Smiling). Rapid progress in generative models, most notably diffusion-based models (Ho et al., 2020; Saharia et al., 2022), provides a clear mitigation method that alleviates bias using synthetic data. For example, Additive Synthetic balancing (ASB) (Ramaswamy et al., 2021) augments the biased real dataset with a balanced synthetic dataset. Uniform Synthetic Balancing (USB) generates enough data to uniformly balance the dataset subgroups (Wang et al., 2020b; Mondal et al., 2023), *i.e.*, each subgroup will have the same number of samples. However, by training the real and synthetic data samples at the same time, a model may simply learn to identify correlations between bias $B$ and whether the data was real or generated $G$ by using generative model artifacts (Corvi et al., 2023). For example, in the setting where the training data contained mostly smiling men but few smiling women, prior work may simply learn that synthetically generated women may smile (but images of real women do not). Thus, as shown in Figure 1(a) and (b), models trained using strategies of prior work (*e.g.*, ASB and USB) may focus on unrelated features for the target task. In addition, we provide theoretical analysis where we prove that every possible augmentation of a biased dataset with synthetic data is going to exhibit some bias toward $(B, G)$. Refer to Section 3.1 for more details.

To mitigate this problem, we rethink how synthetic data is used for bias mitigation by developing a two-stage training pipeline that we call From Fake to Real (FFR). The first step involves pre-training on balanced synthetic data where we learn robust representations across subgroups, thus ensuring,

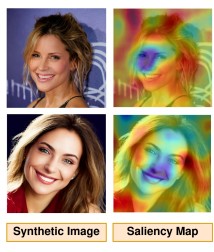 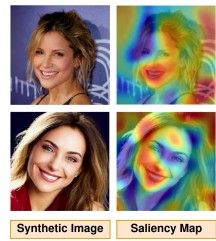 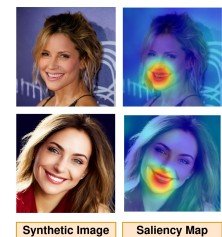

(a) Additive Synthetic Balancing (ASB)  (b) Uniform Synthetic Balancing (USB)  (c) From Fake to Real (FFR) Ours

Figure 1: Saliency maps produced when predicting the attribute Smiling obtained using RISE (Petsiuk et al., 2018). The model for (a) was trained using ASB (Ramaswamy et al., 2021), (b) was trained with USB (Mondal et al., 2023), whereas (c) was trained with our FFR approach. We find that our method ignores background features and, thus, can make better use of synthetic data to mitigate spurious behavior.

$P_D(Y|B) = P_D(Y)$. In the second step, FFR fine-tunes the model on real data using ERM or common loss based bias mitigation methods (Hong & Yang, 2021; Kim et al., 2019; Ryu et al., 2017; Tartaglione et al., 2021; Sagawa et al., 2020a). By separating the two data sources (*i.e.* Real and Synthetic) into two different training steps, FFR avoids the issue of bias that might arise from training on these two sources of data together due to distributional differences between real and synthetic data (*e.g.* generative model artifacts (Corvi et al., 2023)). Effectively, the synthetic data acts as a source of unbiased representations for each subgroup, leading to improved performance when training with the real data using ERM or loss-based bias mitigation methods in the second step. As shown in Figure 1(c), this enables FFR to learn more relevant features rather than focusing on spurious background features.

To evaluate our approach, we expand on the experimental frameworks used in prior work which are limited to one bias rate per dataset (Sagawa et al., 2020b; Qraitem et al., 2023; Joshi et al., 2023). Instead, we conduct systemic analysis over three datasets, CelebA HQ (Lee et al., 2020), UTK-Face (Zhang et al., 2017), and SpuCO Animals (Joshi et al., 2023), and a range of bias rates. We find our approach is especially beneficial for datasets with high levels of bias.

Our contributions are summarized below:

- We introduce a simple, yet effective, two-step training pipeline (FFR) that uses synthetic data to alleviate the issue of spurious correlations (Bias). Our pipeline, unlike prior work, avoids the issue of bias to distributional differences between real-synthetic data (*e.g.* generative model artifacts) and thus is more effective at mitigating bias.
- We provide a theoretical analysis on how augmentation with synthetic data results in an unexpected bias toward synthetic artifacts.
- Comprehensive experiments over three datasets (UTK-face, CelebA HQ, and SpuCO Animals) and four bias strengths per dataset validate our method's effectiveness. Indeed, FFR improves performance over state-of-the-art by 7-13% in high-bias settings.

## 2 RELATED WORK

**Mitigating Bias with Synthetic Data.** As noted in the Introduction, there is some limited work on using synthetic data augmentation for addressing issues stemming from imbalanced training data. This includes Uniform Synthetic Balancing (USB) (Wang et al., 2020b; Mondal et al., 2023), which can be used to balance underrepresented subgroups, where subgroups are the intersection of classes $Y$ and bias groups $B$. This, in turn, effectively ensures that $Y$ is statistically independent from $B$, *i.e.*, $P_{\bar{D}}(Y|B) = P_{\bar{D}}(Y)$ where $\bar{D}$ is the combined dataset of real and synthetic data. Additive Synthetic Balancing (ASB) (Ramaswamy et al., 2021) augments a biased real dataset with a balanced synthetic dataset. In our work, we show how both approaches (USB and ASB) result in models that are biased toward $(B, G)$ where $G = \{Real, Synthetic\}$, *i.e.*, the variable that differentiates between real and synthetic data. We could attempt to mitigate this issue by combining USB and ASB with loss-based bias mitigation methods (*e.g.*, (Hong & Yang, 2021; Kim et al., 2019; Ryu

et al., 2017; Tartaglione et al., 2021; Sagawa et al., 2020a)). However, in order to account for the new source of bias from $(B, G)$ where $G = \{Real, Synthetic\}$, this approach doubles the number of bias groups ($|(B, G)| = |B||G| = 2|B|$) which increases the optimization difficulty, reducing performance. Instead, our two-stage training pipeline addresses the issue of new biases being introduced from using synthetic data by training both real and generated data separately.

**Non-Generative Mitigation Methods.** Also related to our are methods that use architecture changes and/or alters the training procedures to mitigate dataset bias (Ryu et al., 2017; Kim et al., 2019; Wang et al., 2020c; Hong & Yang, 2021; Tartaglione et al., 2021; Sagawa et al., 2020b). For example, Sagawa et al. (2020b) presents GroupDRO (Distributionally Robust Neural Networks for Group Shifts), a regularization procedure that adapts the model optimization according to the worst-performing group. Our work complements these efforts by introducing a novel pipeline for using synthetic data that further boosts the performance of these methods especially in high bias settings.

**Uncovering Spurious Correlations.** In our work, we are interested mitigating spurious correlations; a spurious correlation results from underrepresenting a certain group of samples (*e.g.* samples with the color red) within a certain class (*e.g.* planes) in the training set. This leads the model to learn the false relationship between the class and the over-represented group. Prior work has documented several occurrences of this bias. For example, Singh et al. (2020); Hendrycks et al. (2021); Xiao et al. (2020); Li et al. (2020) showed that state-of-the-art object recognition models are biased toward backgrounds or textures associated with the object class. Agrawal et al. (2018); Clark et al. (2019) showed similar spurious correlations in VQA. Most recently, Meister et al. (2022) demonstrates how biases toward gender are ubiquitous in the COCO and OpenImages datasets. As the authors demonstrate, these artifacts vary from low-level information (e.g., the mean value of the color channels) to higher level (e.g., pose and location of people).

## 3 SYNTHETIC DATA FOR ROBUST REPRESENTATIONS AGAINST BIAS

Visual classification models can often rely on "spurious" correlations encoded in the training set that don't reflect their real-world distributions. More concretely, given a dataset of images $X$, classes $Y$, and bias signal $B$ (*e.g.*, Gender: Male/Female), a biased model relies on the signal in $X$ that infer $B$ to make predictions $\hat{Y}$. This is often because the distribution $P_D(Y|B) \neq P_D(Y)$, *i.e.*, the training set encodes some correlation between the classes and the biases. For example, given a particular class $y$ (*e.g.*, Smiling), a certain bias group $b$ (*e.g.*, Male) might be over-represented when compared to others. Therefore, a model might mistakenly predict the class of an image (*e.g.*, Not Smiling) as the wrong class (*e.g.*, Smiling) because the signal $b$ (Male) is present in the image (*e.g.*, Man is Not Smiling).

To address this issue, our work explores the usage of synthetic data from generative models. In Section 3.1, we explore how augmenting the real dataset with synthetic data results in a bias towards distributional differences between synthetic and real data. In Section 3.2, we introduce From Fake to Real (FFR); novel two-stage pipeline that addresses this issue. Finally, in Section 3.3, we describe our systematic experimental framework and note how it expands on prior work.

### 3.1 MOTIVATION

In this section, we explore a critical problem with the class of solutions that mitigates dataset bias by augmenting biased datasets with synthetic data, *e.g.*, Additive Synthetic Balancing (ASB) (Ramaswamy et al., 2021) and Uniform Synthetic Balancing (USB) (Mondal et al., 2023). These approaches don't consider the fact that the distribution of synthetic data is not the same as the distribution of real data. Indeed, while research on generative models has made significant progress in producing ever more realistic images, especially with the recent advent of diffusion models (Ho et al., 2020; Saharia et al., 2022), there might still be some distributional differences between the real and synthetic data. For example, Corvi et al. (2023) demonstrates how state-of-the-art diffusion models leave fingerprints in the generated images that could be used by recognition models to differentiate between real and synthetic data.

Assuming real and synthetic data are drawn from different distributions, and we are given a biased dataset $D$, *i.e.* $P_D(Y|B) \neq P_D(Y)$, we argue that it is impossible to guarantee that we can create $\bar{D}$ where $Y$ is not biased toward the pair $(\bar{B}, G)$. Formally:

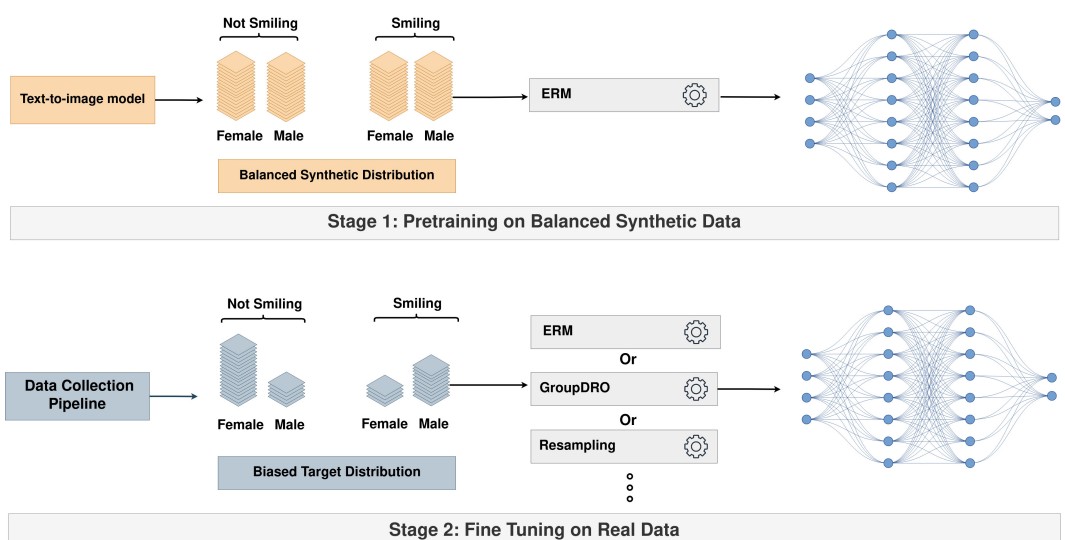

Figure 2: An overview of From Fake to Real (FFR) that incorporates synthetic data to mitigate bias. In Stage 1, we pretrain on a balanced synthetic dataset where we learn robust representations across subgroups. In Stage 2, we fine-tune the model on real data using ERM or common non-data-based bias mitigation methods. By training on real and synthetic data separately, we avoid the issue of bias between the two data sources. Moreover, synthetic data act as an initial source of robust representation that improves the performance of the fine-tuning method in the second step. Refer to Section 3.2 for further discussion.

**Theorem 1.** *Assume we are given dataset $D$ where $P_D(Y|B) \neq P(B)$ such that $Y$ are target labels and $B$ are biased group labels (i.e. dataset is biased). Assume $\bar{\mathcal{D}}$ represent all possible versions of the dataset augmented with synthetic data such that $G = \{Real, Synthetic\}$, then for every $\bar{D} \in \bar{\mathcal{D}}$, $P_{\bar{D}}(Y|B,G) \neq P_{\bar{D}}(Y)$ where $G$ are the synthetic/real labels.*

Refer to Appendix C for proof. Indeed, this Theorem guarantees that it is impossible to create an augmented version of the dataset $D$, *i.e.* $\bar{D}$ without $\bar{D}$ exhibiting some bias toward $(B, G)$. Therefore, this implies that both methods from prior work, ASB (Ramaswamy et al., 2021) and USB (Wang et al., 2020b; Mondal et al., 2023), may rely on signals from $(B, G)$ to make predictions.

To gain some intuition, consider the following illustrative example for Uniform Synthetic Balancing (USB): in an attempt to mitigate the dataset bias of class Landbirds being mostly on Land and Waterbirds being most Water, a significant number of synthetic samples of Landbirds on Water and Waterbirds on Land are added to the dataset. While this means that there is an equal number of Landbirds and Waterbirds on Land and on Water in the combined dataset, *i.e.*, $P_{\bar{D}}(Y|B) = P_{\bar{D}}(Y)$, this also means that there are significantly more *Synthetic* Landbirds on Water than there are *Synthetic* Landbirds on Land. Assuming that the model could differentiate between real and synthetic images, then it is likely advantageous to learn the signal pair (Water, Synthetic) in order to predict the class Landbird while the signal (Water, real) predicts the class Waterbirds.

## 3.2 FROM FAKE TO REAL (FFR): A TWO-STAGE TRAINING PIPELINE

Our approach, From Fake to Real (FFR), aims to address the issue that arises in prior work where models learn a bias between the target labels $Y$ and the pair labels $(B, G)$ as outlined in Section 3.1. The key to our approach is the separation of training on the two data sources, real and synthetic, into two different stages. The model is exposed to one data source at a time, which effectively prevents the use of signals from the pair $(B, G)$ to make predictions as neither appear in the same training step. We provide additional details on our two training stages below:

**Step 1:** FFR pretrains a model $M$ on a balanced synthetic dataset $D_{syn}$ where $P_{D_{syn}}(Y|B) = P_{D_{syn}}(Y)$. To obtain this distribution, we simply deploy a generative model to generate the same number of synthetic data per bias subgroup. This step enables the model $M$ to learn robust initial

representations for each subgroup. Refer to Figure 2 (Stage 1) for an overview of this step. Denote the resulting model from this step as $\bar{M}$.

**Step 2:** While Step 1 learns valuable unbiased representations, there are still a distribution shift going from real to synthetic datasets (Sariyildiz et al., 2023). Therefore, we fine-tune the model from Step 1, *i.e.* $\bar{M}$ on the real dataset to better fit to its distribution. We find that even a simple empirical-risk minimization fine-tuning using the model $\bar{M}$ as an initialization is sufficient to boost performance. However, the real dataset's distribution $D$ is biased, *i.e.*, $P_D(Y|B) = P_D(Y)$. Thus, some of the benefit of our first stage pretraining is undone as the model might simply relearn the bias. To address this, we combine our two stage training pipeline with loss-based bias mitigation methods (*e.g.*, (Hong & Yang, 2021; Kim et al., 2019; Ryu et al., 2017; Tartaglione et al., 2021; Sagawa et al., 2020a)). Refer to Figure 2 (Stage 2) for an overview of this step. As we note in our experiments, regardless of the method used in Step 2, we note a significant performance boost using Step 1's model $\bar{M}$ for initialization.

In a nutshell, note how our method is a flexible framework that rethinks the use of synthetic data for bias mitigation. Indeed, our framework deploys synthetic data to learn initial unbiased representations to improve the performance of training on real data regardless of the method used to train on real data. Therefore, it is generalizable to any bias mitigation method and easy to implement no matter the model architecture. Finally, our framework effectively avoids the issue of bias to distributional differences between real and synthetic data, unlike prior work methods.

### 3.3 SYSTEMIC ANALYSIS OF THE EFFECTIVENESS OF SYNTHETIC DATA IN BIAS MITIGATION

We hypothesize our method is most effective in high-bias settings where the majority subgroups constitute most of the class's data. In these settings, we expect loss-based bias mitigation methods to struggle in alleviating the bias using the very few samples in the minority subgroups. To verify this claim, we expand on the experimental analysis of prior work bias mitigation efforts (Sagawa et al., 2020b; Qraitem et al., 2023; Joshi et al., 2023) by considering a wider range of bias rates per dataset. Indeed, we create four splits of each dataset where each split reflects a bias that ranges from moderate to severe. We achieve this by simply dropping samples from each dataset until the majority groups in each class represent $x\%$ of the class where $x$ denotes the "bias level".

## 4 EXPERIMENTS

**Datasets** We use three datasets 1) CelebA HQ (Lee et al., 2020) where we use the gender attribute as the bias variable and smiling as the target attribute 2) UTK-Face dataset (Zhang et al., 2017) where we use age as the bias attribute and gender as the target attribute. 3) we use the recently introduced SpuCO Animals dataset (Joshi et al., 2023) specifically designed to test spurious correlations where the bias attributes are {*Indoors, Outdoors, Land, Water*} and target attributes are {*Small dogs, Big Dogs, Landbirds, Waterbirds* }. Moreover, for each dataset, we train on 4 different splits where we vary the bias of the majority groups between {95%, 97%, 99%, 99.9%}.

**Metrics** Following Sagawa et al. (2020b), we use Worst Accuracy (WA) to measure the models' spurious behavior. This metric returns the accuracy of the worst performing subgroup where the subgroup is defined as the intersection of class and bias group. In addition, we use balanced accuracy (BA) which averages the accuracies of all subgroups (Qraitem et al., 2023). BA reflects the overall performance of the model while not being biased by the majority subgroups.

**Implementation Details** We train Resnet50 (He et al., 2016) models on every datasets. For optimization, we use ADAM optimizer (Kingma & Ba, 2015) where we grid search the learning rate over the validation set. We use default values for the other parameters. We do not use any learning rate scheduler or augmentations. Refer to the Appendix A for the exact choices of each method. For generation, we use Stable Diffusion V1.4 (Rombach et al., 2022) where we use the prompt template *A photo of* {*bias*} {*class*}.

**Baselines** We report the performance of training with Empirical Risk Minimization (ERM) and several popular state-of-the-art bias mitigation methods. Denote the set of these methods as $X$, we report the performance of four variations of each $x \in X$:

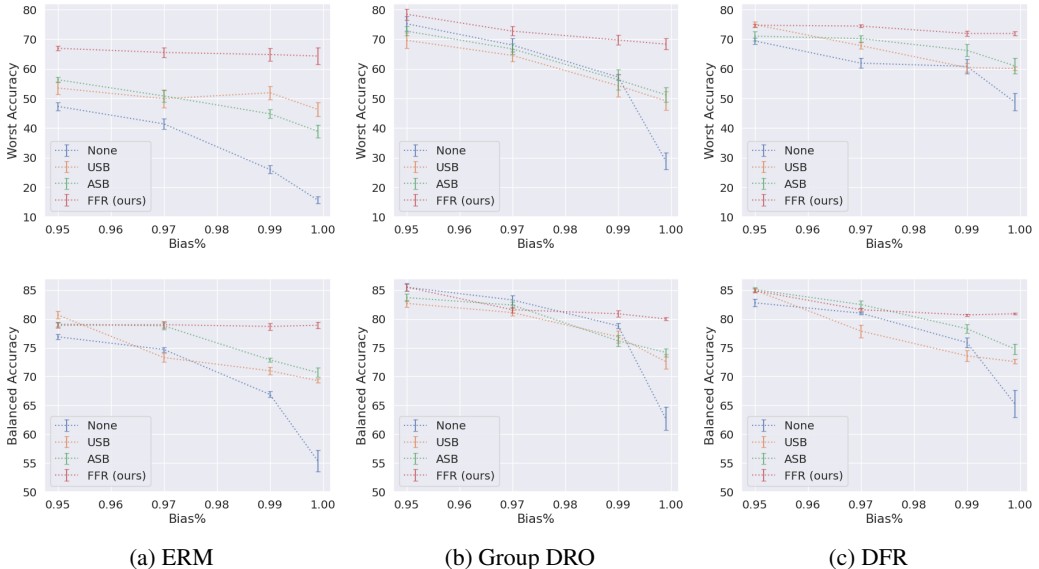

|          |          |          |
|:--------:|:--------:|:--------:|
| (a) ERM | (b) Group DRO | (c) DFR |

Figure 3: Comparison of performance averaged over three datasets: UTK-Face (Zhang et al., 2017), CelebA HQ (Lee et al., 2020), and SpuCO Animals (Joshi et al., 2023) between the effect of three different modes of training: (None) no synthetic data is used, (USB) synthetic data is used to uniformly balance the distribution (extension of prior work on imbalanced classification (Mondal et al., 2023)), (ASB) balanced synthetic data is added to the real dataset (ASB) (Ramaswamy et al., 2021) and (FFR) our method where pretrain on balanced synthetic data and fine tune on real data on three different training algorithms: ERM, Group DRO (Sagawa et al., 2020b) and DFR Kirichenko et al. (2023). Refer to Section 4.1 for discussion.

- No synthetic data is used (None)
- Synthetic data is used to uniformly balance the distribution (USB)
- A balanced synthetic dataset is added to the real dataset (ASB)
- Our method (FFR) where we first pre-train on balanced synthetic data using ERM and then fine-tune on each $x \in X$

With respect to the bias mitigation methods, we report the performance of Group DRO (Sagawa et al., 2020b), Resampling, and Deep Feature Reweighting (DFR) (Kirichenko et al., 2023). Group DRO is an optimization technique where the contribution of each subgroup loss is weighted by their performance. Resampling oversamples minority subgroups such that each subgroup is equally represented per batch. DFR trains a model with ERM and then trains a linear layer over the feature space on a balanced validation set.

## 4.1 SYSTEMIC ANALYSIS OF SYNTHETIC DATA USAGE ACROSS SEVERAL BIAS RATIOS

To gain a holistic understanding of the methods' performance, we average our results over the three datasets and report them in Figure 3. Due to space constraints, we provide the results of Resampling in the Appendix. First, note that *FFR overall is better than ASB and USB at improving performance, regardless of the method used to train the model (*i.e. *ERM, DFR, GroupDRO), especially at high bias settings*. This is evident from the large gap in worst accuracy (top row) on bias settings $\{99\%, 99.9\%\}$. This is likely because, as we discussed in 3.2, our method is more effective at fixing the issue of bias between real and synthetic data.

For individual methods we find FFR obtains the highest gains using ERM. This is likely because, as we discussed in 3.2, our method effectively fixes the issue of bias between real and synthetic data without the intervention of any loss-based bias mitigation methods. Moreover, even though our method, in its second step, fine-tunes the model on the real biased distribution, the model doesn't fully relearn the bias and is able to outperform USB and ASB. This is likely because the robust representations from the first step make it easier and, thus, more advantageous to learn generalizable

| Method | Synthetic Data Usage | Utk-Face | | CelebA HQ | | SpuCo Animals | |
|---|---|---|---|---|---|---|---|
| | | WA | BA | WA | BA | WA | BA |
| (a) ERM | None | 5.9 | 51.3 | 37.6 | 57.3 | 3.7 | 57.6 |
| | USB | 33.0 | 56.8 | 51.3 | 72.2 | 54.7 | 78.9 |
| | ASB | 23.2 | 60.1 | 57.3 | 78.0 | 39.9 | 73.4 |
| | FFR (ours) | 42.9 | 65.3 | 77.1 | 86.8 | 74.9 | 84.5 |
| (b) Resampling | None | 29.9 | 56.4 | 44.4 | 63.9 | 44.0 | 75.3 |
| | USB | 31.2 | 58.5 | 57.4 | 70.0 | 69.9 | 80.0 |
| | ASB | 31.1 | 59.2 | 54.0 | 73.6 | 62.1 | 80.2 |
| | FFR (ours) | 49.3 | 69.0 | **86.4** | **89.7** | 74.5 | 84.4 |
| (c) GroupDRO (Sagawa et al., 2020b) | None | 23.3 | 55.1 | 55.2 | 80.1 | 8.4 | 53.0 |
| | USB | 34.6 | 59.7 | 66.2 | 83.7 | 48.5 | 74.3 |
| | ASB | 42.3 | 62.2 | 60.0 | 81.4 | 51.4 | 79.1 |
| | FFR (ours) | 47.2 | 66.4 | 83.3 | 89.3 | 75.1 | 84.3 |
| (d) DFR (Kirichenko et al., 2023) | None | 35.2 | 59.4 | 57.6 | 61.4 | 53.7 | 75.0 |
| | USB | 45.8 | 56.5 | 64.5 | 76.2 | 69.8 | 84.9 |
| | ASB | 46.8 | 59.2 | 72.3 | 80.8 | 67.7 | 84.5 |
| | FFR (ours) | **55.3** | **69.2** | 85.1 | 88.3 | **75.4** | **85.3** |

Table 1: Comparison over three datasets: UTK-Face (Zhang et al., 2017), CelebA HQ (Lee et al., 2020), and SpuCO Animals (Joshi et al., 2023) between the impact of four synthetic data augmentation methods: no synthetic data is used, synthetic data is used to uniformly balance the distribution (USB) ((Mondal et al., 2023)), balanced synthetic data is added to the real dataset (ASB) (Ramaswamy et al., 2021) and FFR (ours) (ours) where the model is pretrained on balanced synthtetic data. We study the effect of each method on the performance of ERM and several bias mitigation methods (Group DRO, Resampling, and DFR). **Bolded** Numbers reflect best performance across methods. Underlined numbers reflects the best performance per method. Note how the best performance is achieved when our method (FFR) is used. Refer to Section 4.2 for discussion.

features than relearning the bias. Nevertheless, note that prior work synthetic augmentation methods (USB and ASB) also manage to improve the performance of ERM over each bias ratio. This indicates that, indeed, synthetic data augmentation is able to alleviate bias as noted in (Ramaswamy et al., 2021). Overall, note that these improvements (From USB, ASB and our method FFR) are most significant in a high bias ratio, namely, $\{99\%, 99.9\%\}$ where augmenting with real data is really helpful for learning good generalizable representations.

We now shift our focus to Figure 3(b) and (c), which use GroupDRO (Sagawa et al., 2020b) and DFR (Kirichenko et al., 2023), respectively. We find our method (FFR) obtains best performance, especially in high bias settings. Note, however, how the use of synthetic data through either USB, ASB, or FFR doesn't meaningfully impact the performance of these methods over moderate amounts of bias, *e.g.*, $\{95\%, 97\%\}$. This means that at these bias ratios, DFR and GroupDRO are robust enough to alleviate the bias without the intervention of synthetic data. Thus, this suggest that the high bias setting is where synthetic data is most helpful.

## 4.2 A CLOSER LOOK INTO A CHALLENGING HIGH-BIAS SETTINGS

In Table 1, we report performance when majority subgroups represent $99.9\%$ of their respective class's data. We make three major observations from this data. First, methods that use FFR obtain the best performance overall, namely combining FFR with Resampling on CelebA HQ (Table 1(c)) and combining FFR with DFR for UTK-Face and SpuCo Animals (Table 1(d). Overall, note that our method improves worst group accuracy by $7 - 14\%$ when compared to the next best synthetic data augmentation method for each dataset. In addition, each individual method second stage mitigation method (ERM, Resampling, GroupDRO, and DFR) is improved by first using our first stage pretraining method. This demonstrates that our FFR method generalizes across datasets and bias mitigation methods.

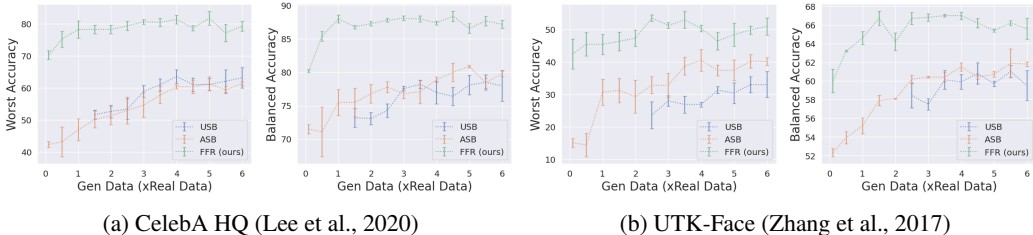

(a) CelebA HQ (Lee et al., 2020)  (b) UTK-Face (Zhang et al., 2017)

Figure 4: Comparison over (a) CelebA HQ and (b) UTK-Face between the Synthetic Augmentation nethods: Uniform Synthetic Balancing (USB) (Mondal et al., 2023), Additive Synthetic Balancing (ASB) (Ramaswamy et al., 2021) and ours (FFR) where synthetic data is scaled up to 6 times the size of each dataset. Note how our method consistently performs better than USB and ASB even as data is scaled. Refer to Section 4.3 for further discussion.

As we discuss in the Introduction, one could improve USB and ASB by combining them with bias mitigation methods (*i.e.* GroupDRO, Resampling, and DFR). As shown in Table 1, they do obtain a performance boost but perform worse than our FFR method. This is likely due, in part, to the fact FFR automatically mitigates and resolves the issue of bias to distributional differences (*e.g.* synthetic artifacts) via our two stage training process. In addition, combining USB and ASB is more challenging as both the bias and the data source (synthetic vs. real) must be taken into account for methods like GroupDRO and Resampling, resulting in double the number of bias subgroups.

### 4.3 Effect of Scaling Synthetic Data on Performance

In Section 4.1 and Section 4.2, we fixed the size of synthetic data used during training according to the constraint of Unfiorm Synthetic Balancing (USB). In this Section, we relax this constraint; we study the effect of scaling synthetic data on prior work methods: Uniform Synthetic Balancing (USB), Additive Synthetic Balancing (ASB), and our method From Fake to Real (FFR), each trained with ERM. We perform this analysis on CelebA HQ (Lee et al., 2020) and UTK-Face (Zhang et al., 2017) where we scale synthetic data up to 6 times the size of each dataset. Figure 5 reports our results, where we find our method (FFR) continues to achieve the best performance over both datasets even as we scale the data. This is likely because our method effectively addresses the issue of bias to distributional differences between real and synthetic data. Moreover, we report that our method performance plateaus at an earlier than either USB or ASB. Thus, this shows our method is more data efficient while also achieving the best results.

### 4.4 Qualitative analysis

In this Section, we conduct a qualitative comparison between ERM without any synethic data, Additive Synthetic Balancing (ASB), Uniform Synthetic Balancing (USB), and our method From Fake to Real (FFR) on the SpuCO Animals dataset (Joshi et al., 2023) with bias rate 99.9%. Note that the dataset contains four classes: Big Dogs, Small Dogs, Landbirds, and Waterbirds. In this section, we focus on the minority subgroups "Big Dogs Indoors" and "Small Dogs outdoors" and sample a real and synthetic image from each subgroup. For each image and model, we produce a saliency map using RISE (Petsiuk et al., 2018). Figure 5 reports our results, where we find FFR is the only method that is able to focus on the dog features while disregarding features from the background in both the synthetic and real images. For example, in the second row, both ASB and USB pay attention to the man's feet as well as the ground floor and what seems the bottom of a couch to make predictions. Whereas our method (FFR) only focuses on the dog features. More interestingly, note how for the synthetic images in rows 1 and 3, prior work methods (ASB and USB) use generative artifacts (*e.g.* three "toes" for the dog rather than four) to make predictions whereas our method (FFR) ignores these features. This helps validate our hypothesis that our method is effective at resolving the issue of bias toward the distributional differences between real and synthetic data.

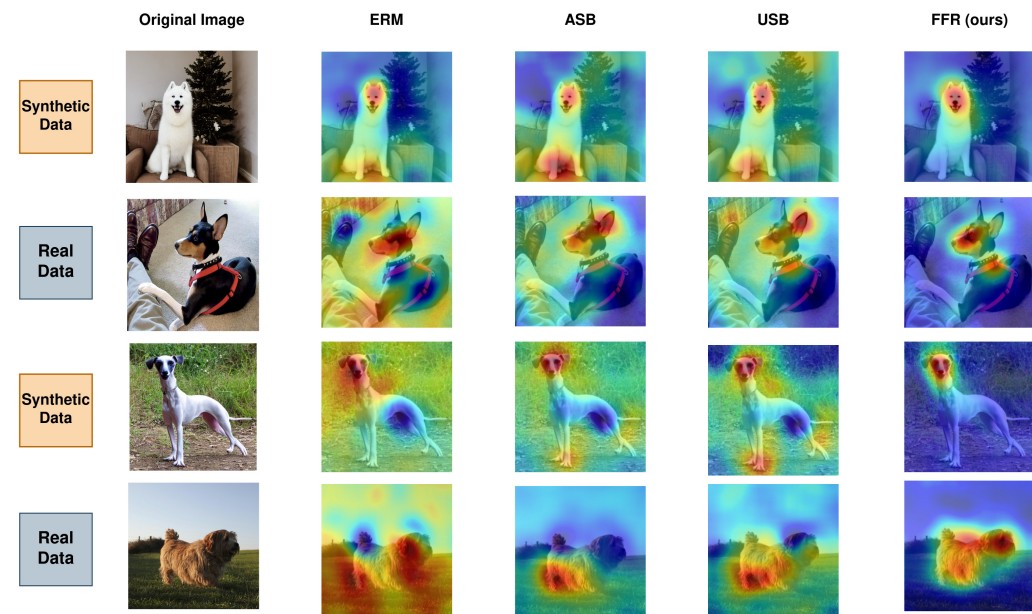

Figure 5: Saliency maps produced when predicting Big Dogs (top two rows) and Small dogs (bottom two rows) obtained using RISE (Petsiuk et al., 2018) using ERM, ASB (Ramaswamy et al., 2021), USB (Wang et al., 2020b; Mondal et al., 2023) and our method FFR to augment the dataset with synthetic data. The real images are from the dataset SpuCO Animals Joshi et al. (2023) and the synthetic data is from Stable Diffusion v1.4 Rombach et al. (2022). Note how our method (FFR) is the only method that is able to localize the relevant features (dog features) and not get distracted by spurious background features. Refer to Section 4.4 for discussion.

## 5 CONCLUSION

In this work, we investigated the use of synthetic data augmentation for bias mitigation. We provided a theoretical argument that demonstrates how methods that augment biased datasets with synthetic data do not fully mitigate the original dataset bias (Ramaswamy et al., 2021). In fact, they result in a new bias that is a function of the original data bias as well as synthetic data artifacts. To address these issues, we rethought the way biased data is used for bias mitigation by developing a two-step pipeline that we called From Fake to Real (FFR). The pipeline separates training on synthetic data from training on real data, thus, avoiding the bias toward synthetic artifacts. As a result, synthetic data is used as an initial source of robust representations that improve the performance of training on biased real data regardless of the method used (*e.g.* ERM, GroupDRO (Sagawa et al., 2020b), DFR (Kirichenko et al., 2023), etc). Our systemic analysis over three datasets and four bias settings per dataset demonstrated how our method is more effective at using synthetic data than prior work methods and thus undoing the bias. Indeed, FFR outperforms prior work state-of-the-art by $7-14\%$ on high bias settings. We then demonstrated how our method is more data efficient; it requires less synthetic data to achieve the best performance. Finally, we provided an extensive qualitative analysis using saliency maps where we demonstrated how our method is able to disregard spurious features, unlike other synthetic augmentation methods.

**Limitations and Future Work** As discussed in our work, we use large pre-trained text-to-image models to generate synthetic data. While the property of controllable generation using text allows us to generate data that undoes the bias of the real dataset, the generative model might nevertheless inject some biases into the generated data that are not accounted for by the text used to generate. For example, Stable Diffusion (Rombach et al., 2022) used in this work has been demonstrated to exhibit several biases (Luccioni et al., 2023; Bianchi et al., 2023). Therefore, future research that focuses on training fairer generative models would alleviate some of these issues. Moreover, even though we make sure to use the latest datasets for benchmarking bias mitigation methods, we note that these datasets are nevertheless smaller than modern datasets used to train large-scale recognition systems. Therefore, future work could benefit from collecting larger datasets to evaluate the robustness of bias mitigation methods.

## 6   CODE OF ETHICS STATEMENT

Our work addresses a critical problem with recognition models: spurious predictive behavior. We measure this behavior by calculating the accuracy of dataset's subgroups. While this metric aligns with our goal of preventing spurious behavior, we emphasize that the metric is not exhaustive of other fairness concerns. We refer the reader to (Kleinberg et al., 2017) for a broader discussion of fairness metrics. Additionally, though our suggested approach seeks to learn resilient representations for minority subgroups within a given dataset, we recognize the potential for these representations to be inappropriately employed in subsequent applications (*e.g.* surveillance).

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

## A  HYPERPARAMETERS

For experiments in Sections 4.1 and 4.2, we provide the learning rates used to train our models for each dataset in Table 2 following a grid search over the validation set. Note that for weight decay, we use $1e - 05$ for UTK-Face, CelebA-HQ and $1e - 03$ for SpuCO Animals following Joshi et al. (2023). For the experiment in Section 4.3, we use the same learning rate from Table 2 across the different data scales.

## B  SYNTHETIC DATA AUGMENTATION EFFECT ON RESAMPLING

In Section 4.1, we omit the results of Resampling combined with Synthetic Data Augmentation methods due to space constraints. Observe the results here in Figure 6. Similar to the discussion with respect to GroupDro and DFR in Section 4.1, we note that our method (FFR) is better than Uniform Synthetic Balancing (USB) and Additive Synthetic Balancing (ASB) at improving resampling performance on high bias settings, *i.e.*, $\{97\%, 99\%, 99.9\%\}$. On moderate bias settings, *i.e.*, $\{95\%\}$, we note that Resampling without synthetic data is sufficient to achieve the best performance.

## C  PROOFS

We first prove the following helpful Lemma:

**Lemma 1.** *Assume that $P_D(Y|B) = P_D(Y)$, then for any $y, y' \in Y$ and $b \in B$, we get $P_D(B = b|Y = y) = P_D(B = b|Y = y')$*

*Proof.* Given any $b \in B$:

$$P_D(B = b|Y = y) = \frac{P_D(Y = y|B = b)P_D(B = b)}{P_D(Y = y)} \tag{1}$$

$$= \frac{P_D(Y = y)P_D(B)}{P_D(Y = y)} \tag{2}$$

$$= P_D(B = b) \tag{3}$$

Note that (2) simply follows by definition of $P_D(Y|B) = P_D(Y)$. Similarly:

$$P_D(B = b|Y = y') = \frac{P_D(Y = y'|B = b)P_D(B = b)}{P_D(Y = y')} = \frac{P_D(Y = y')P(B)}{P_D(Y = y')} = P_D(B = b) \tag{4}$$

Thus,

$$P_D(B = b|Y = y) = P_D(B = b) = P_D(B = b|Y = y') \tag{5}$$

$\square$

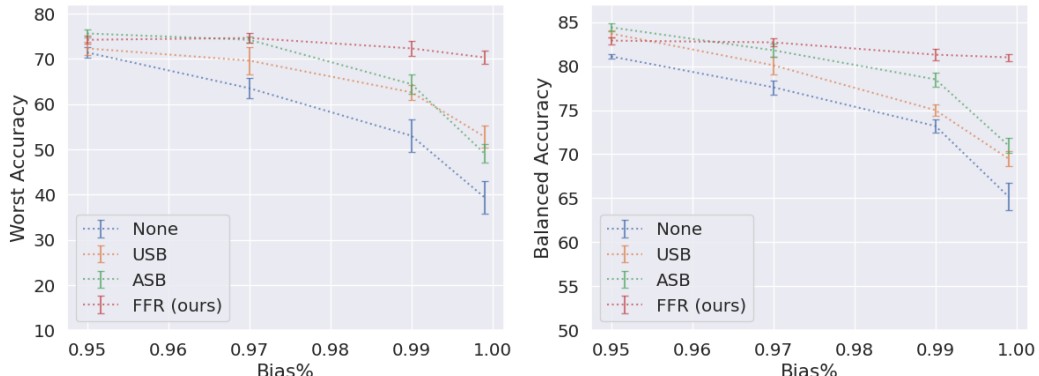

Figure 6: Comparison of performance averaged over three datasets: UTK-Face (Zhang et al., 2017), CelebA HQ (Lee et al., 2020), and SpuCO Animals (Joshi et al., 2023) between the effect of three different modes of training: (None) no synthetic data is used, (USB) synthetic data is used to uniformly balance the distribution (extension of prior work on imbalanced classification (Mondal et al., 2023)), (ASB) balanced synthetic data is added to the real dataset (ASB) (Ramaswamy et al., 2021) and (FFR) our method where pretrain on balanced synthetic data and fine tune on real data on Resampling. Refer to Appendix B for discussion

Now, we proceed to prove Theorem 1:

**Theorem 1.** *Assume we are given dataset $D$ where $P_D(Y|B) \neq P(B)$ such that $Y$ are target labels and $B$ are biased group labels (i.e. dataset is biased). Assume $\bar{\mathcal{D}}$ represent all possible versions of the dataset augmented with synthetic data such that $G = \{Real, Synthetic\}$, then for every $\bar{D} \in \bar{\mathcal{D}}$, $P_{\bar{D}}(Y|B,G) \neq P_{\bar{D}}(Y)$ where $G$ are the synthetic/real labels.*

*Proof.* We will prove this by contradiction. Assume that $P_{\bar{D}}(Y|B,G) = P_{\bar{D}}(Y)$. By Definition of the biased dataset, there exists $b, b' \in B$ and $y, y' \in Y$ such that

1. $P_D(B = b|Y = y) > P_D(B = b'|Y = y)$

2. $P_D(B = b'|Y = y') > P_D(B = b|Y = y')$

Now, assume $Count_D(Y = y, B = b, G = g)$ is an operator that returns the number of samples given class $y$, bias $b$ and real/synthetic label $g$ in dataset $D$. Moreover, denote the following variables:

1. $M = Count_{\bar{D}}(B = b, Y = y, G = real)$

2. $N = Count_{\bar{D}}(B = b', Y = y, G = real)$

3. $M' = Count_{\bar{D}}(B = b, Y = y', G = real)$

4. $N' = Count_{\bar{D}}(B = b', Y = y', G = real)$

Then it follows that given any $\bar{D} \in \bar{\mathcal{D}}$, then:

1. $M > N$

2. $N' > M'$

Now, observe:

$$P_{\bar{D}}(B = b, G = real | Y = y) = \frac{M}{\sum_b Count_{\bar{D}}(B = b, Y = y, G = Real) \atop + \sum_b Count_{\bar{D}}(B = b, Y = y, G = Synthetic)} \qquad (6)$$

$$> \frac{N}{\sum_b Count_{\bar{D}}(B = b, Y = y, G = Real) \atop + \sum_b Count_{\bar{D}}(B = b, Y = y, G = Synthetic)} \qquad (7)$$

$$= P_{\bar{D}}(B = b', G = real | Y = y) \qquad (8)$$

Similarly:

$$P_{\bar{D}}(B = b', G = real | Y = y') = \frac{N'}{\sum_b Count_{\bar{D}}(B = b, Y = y', G = Real) \atop + \sum_b Count_{\bar{D}}(B = b, Y = y', G = Synthetic)} \qquad (9)$$

$$> \frac{M'}{\sum_b Count_{\bar{D}}(B = b, Y = y', G = Real) \atop + \sum_b Count_{\bar{D}}(B = b, Y = y', G = Synthetic)} \qquad (10)$$

$$= P_{\bar{D}}(B = b, G = real | Y = y') \qquad (11)$$

In order to satisfy the main assumption in our proof, *i.e.*, $P_{\bar{D}}(Y|B, G) = P_{\bar{D}}(Y)$, then following the contrapositive of Lemma 1:

$$P_{\bar{D}}(B = b', G = real | Y = y') = P_{\bar{D}}(B = b', G = real | Y = y) \qquad (12)$$

$$P_{\bar{D}}(B = b, G = real | Y = y') = P_{\bar{D}}(B = b, G = real | Y = y) \qquad (13)$$

To that end, we can change the term: $\sum_b Count_{\bar{D}}(B = b, Y = y, G = Synthetic)$ by adding more synthetic data to the dataset. We can't change $\sum_b Count_{\bar{D}}(B = b, Y = y, G = Real)$ because we don't have access to more real data.

Therefore, according to the results in (8) and (11), adding more synthetic data to achieve (12) implies that:

$$P_{\bar{D}}(B = b, G = real | Y = y) > P_{\bar{D}}(B = b, G = real | Y = y') \qquad (14)$$

which breaks (13). Similarly, achieving (13) by adding more synthetic data implies that:

$$P_{\bar{D}}(B = b', G = real | Y = y') > P_{\bar{D}}(B = b', G = real | Y = y) \qquad (15)$$

which breaks (12). Thus, arriving to a contradiction.

$\square$

|       |                | 95%     | 97%     | 99%     | 99.9%   |
|-------|----------------|---------|---------|---------|---------|
| None  | ERM            | 1.0e-03 | 1.0e-03 | 1.0e-04 | 1.0e-02 |
|       | G-DRO          | 1.0e-03 | 1.0e-03 | 1.0e-03 | 1.0e-04 |
|       | Resampling     | 1.0e-02 | 1.0e-02 | 1.0e-03 | 1.0e-02 |
| USB   | ERM            | 1.0e-03 | 1.0e-02 | 1.0e-03 | 1.0e-03 |
|       | G-DRO          | 1.0e-03 | 1.0e-02 | 1.0e-03 | 1.0e-03 |
|       | Resampling     | 1.0e-03 | 1.0e-02 | 1.0e-07 | 1.0e-07 |
| ASB   | ERM            | 1.0e-03 | 1.0e-03 | 1.0e-03 | 1.0e-02 |
|       | G-DRO          | 1.0e-03 | 1.0e-02 | 1.0e-03 | 1.0e-03 |
|       | Resampling     | 1.0e-02 | 1.0e-07 | 1.0e-07 | 1.0e-07 |
| FFR   | ERM (Pretrain) | 1.0e-05 | 1.0e-04 | 1.0e-04 | 1.0e-04 |
|       | ERM            | 1.0e-07 | 1.0e-07 | 1.0e-07 | 1.0e-07 |
|       | G-DRO          | 1.0e-03 | 1.0e-03 | 1.0e-07 | 1.0e-07 |
|       | Resampling     | 1.0e-07 | 1.0e-06 | 1.0e-07 | 1.0e-07 |

(a) CelebA HQ

|       |                | 95%     | 97%     | 99%     | 99.9%   |
|-------|----------------|---------|---------|---------|---------|
| None  | ERM            | 1.0e-03 | 1.0e-03 | 1.0e-03 | 1.0e-03 |
|       | G-DRO          | 1.0e-04 | 1.0e-04 | 1.0e-04 | 1.0e-03 |
|       | Resampling     | 1.0e-02 | 1.0e-02 | 1.0e-02 | 1.0e-02 |
| USB   | ERM            | 1.0e-04 | 1.0e-02 | 1.0e-02 | 1.0e-02 |
|       | G-DRO          | 1.0e-05 | 1.0e-04 | 1.0e-02 | 1.0e-02 |
|       | Resampling     | 1.0e-03 | 1.0e-02 | 1.0e-02 | 1.0e-02 |
| ASB   | ERM            | 1.0e-03 | 1.0e-04 | 1.0e-02 | 1.0e-02 |
|       | G-DRO          | 1.0e-04 | 1.0e-04 | 1.0e-03 | 1.0e-02 |
|       | Resampling     | 1.0e-06 | 1.0e-06 | 1.0e-06 | 1.0e-02 |
| FFR   | ERM (Pretrain) | 1.0e-04 | 1.0e-04 | 1.0e-04 | 1.0e-04 |
|       | ERM            | 1.0e-05 | 1.0e-06 | 1.0e-06 | 1.0e-06 |
|       | G-DRO          | 1.0e-04 | 1.0e-06 | 1.0e-06 | 1.0e-06 |
|       | Resampling     | 1.0e-05 | 1.0e-06 | 1.0e-06 | 1.0e-06 |

(b) UTK-Face

|       |                | 95%     | 97%     | 99%     | 99.9%   |
|-------|----------------|---------|---------|---------|---------|
| None  | ERM            | 1.0e-05 | 1.0e-05 | 1.0e-04 | 1.0e-04 |
|       | G-DRO          | 1.0e-05 | 1.0e-05 | 1.0e-04 | 1.0e-04 |
|       | Resampling     | 1.0e-05 | 1.0e-05 | 1.0e-04 | 1.0e-04 |
| USB   | ERM            | 1.0e-05 | 1.0e-07 | 1.0e-07 | 1.0e-07 |
|       | G-DRO          | 1.0e-07 | 1.0e-07 | 1.0e-07 | 1.0e-07 |
|       | Resampling     | 1.0e-07 | 1.0e-07 | 1.0e-07 | 1.0e-07 |
| ASB   | ERM            | 1.0e-06 | 1.0e-05 | 1.0e-05 | 1.0e-07 |
|       | G-DRO          | 1.0e-07 | 1.0e-06 | 1.0e-07 | 1.0e-07 |
|       | Resampling     | 1.0e-07 | 1.0e-07 | 1.0e-07 | 1.0e-07 |
| FFR   | ERM (Pretrain) | 1.0e-06 | 1.0e-05 | 1.0e-05 | 1.0e-05 |
|       | ERM            | 1.0e-07 | 1.0e-07 | 1.0e-07 | 1.0e-07 |
|       | G-DRO          | 1.0e-05 | 1.0e-07 | 1.0e-07 | 1.0e-07 |
|       | Resampling     | 1.0e-06 | 1.0e-07 | 1.0e-07 | 1.0e-07 |

(c) SpuCO Animals

Table 2: Learning Rates forexperiments in Sections 4.1 and 4.2,. Refer to Appendix A for further details.

