# OpenReview forum: "From Fake to Real: Pretraining on Balanced Synthetic Images to Prevent Bias"
_ICLR.cc/2024/Conference — ICLR 2024 Conference Withdrawn Submission_

### Official Review · Reviewer_EAfJ · 2023-10-27

**Soundness:** 2 fair
**Presentation:** 1 poor
**Contribution:** 1 poor
**Rating:** 3
**Confidence:** 4

**Summary:**

The paper's objective is to improve the generative model-based de-biasing approach. The rationale behind this effort is rooted in the idea that the model's capacity to distinguish between 'real' and 'fake' images can be seen as another form of bias concept. To achieve this, the paper introduces a two-step training pipeline: initially, pre-training using balanced synthetic data, and subsequently, fine-tuning with real data.

**Strengths:**

- The rationale is commendable.
- The performance has shown a remarkable improvement.

**Weaknesses:**

- The approach is rather simplistic, which may affect the significance of its contribution.
- While there is some theoretical support for the notion that the model could learn to distinguish 'real or fake,' a quantitative analysis is necessary. This could involve assessing the model's performance on a discriminator, for instance.
- The notations employed in the script do not adhere to a professional standard, leading to potential confusion in interpretation. E.g., the notation $P_D(Y|B) \neq P(B)$ is unclear because, in practice, most conditional and marginal probabilities differ, making it challenging to discern the author's intention. Additionally, using $Y$ to represent labels is not accurate, as in a classification task, it typically employs a vector notation, such as $\textbf{y}$.
- The overall presentation would benefit from being more concise and streamlined.

**Questions:**

- How is the balancing of synthetic data achieved? Does this approach assume prior knowledge of the specific biases present in the dataset?
- It seems that the phenomenon of 'catastrophic forgetting' [1] in continual learning could potentially occur when transitioning from step 1 to step 2 if all model parameters are fine-tuned. Could you please clarify which parameters are trainable at each step?

[1] Kirkpatrick, James, et al. "Overcoming catastrophic forgetting in neural networks." Proceedings of the national academy of sciences 114.13 (2017): 3521-3526.

---

> ### Author Response · Authors · 2023-11-17
>
> >The approach is rather simplistic, which may affect the significance of its contribution.
>  We agree with the reviewer that the approach is simple. However, it addresses an issue not solved in prior work and is effective, which is a strength, not a weakness. Moreover, our contribution also involves uncovering a critical issue in using synthetic data for bias mitigation and providing a theoretical analysis in addition to the proposed method that fixes the issue.
> >While there is some theoretical support for the notion that the model could learn to distinguish 'real or fake,' a quantitative analysis is necessary. This could involve assessing the model's performance on a discriminator, for instance.
>
> We provide further insight into this phenomenon by providing the worst group accuracy over the synthetic data for our benchmarked synthetic augmentation methods (USB and ASB) and our method (FFR). Indeed, if the model leveraged such artifact-based correlations, then the model will be biased not only on the real data but also on the synthetic data too. Observe the results in the table below:
>
> |              | Real Data |      | Synthetic Data |      |
> |--------------|:---------:|------|:--------------:|------|
> |              | WA        | BA   | WA             | BA   |
> | USB        | 54.7      | 78.9 | 89.3           | 83.1 |
> | ASB        | 39.9      | 73.4 | 68.7           | 85.2 |
> | FFR (ours) | 74.9      | 84.5 | 94.5           | 97.8 |
>
> Note how both USB and ASB perform poorly on both the real and synthetic data. This indicates that rather than using synthetic data to mitigate the bias and generalize to real data, both methods learned the bias on both real and synthetic data. However, our method (FFR), doesn't suffer from this issue. This is clear from achieving the best worst-group accuracy on both the synthetic and real data.
>
> >The notations employed in the script do not adhere to a professional standard, leading to potential confusion in interpretation.
>
> We thank the reviewer for catching the mistake with $$P_{D}(Y|B) \neq P(B)$$. We believe that this is a typo as all the rest of our notation, in fact, correctly references the dataset on which the probability is calculated. We will fix this in the final version.
>
> >How is the balancing of synthetic data achieved?
>
> As discussed in our work, we ensure that every group in the synthetic pretraining dataset has the same number of samples. And yes, we require that the practitioner is aware of which biases they seek to address.
>
> >It seems that the phenomenon of 'catastrophic forgetting' [1] in continual learning could potentially occur when transitioning from step 1 to step 2 if all model parameters are fine-tuned.
>
> We train the entire model and we note significant improvements from steps 1 to 2 as we can see from the table below, which indicates that our model doesn’t suffer from catastrophic forgetting.
>
> |                                            	| WA                 	| BA                 	|
> |------------------------------------------------|------------------------|------------------------|
> | Stage 1                                  	| 54.3               	| 78.7               |
> | Stage 2                                  	| 3.7                  	| 57.6               |
> | Stage 1 --> Stage 2: FFR (ours)       | 74.9           	|  84.5              |

---

### Official Review · Reviewer_tzaP · 2023-10-29

**Soundness:** 2 fair
**Presentation:** 3 good
**Contribution:** 2 fair
**Rating:** 3
**Confidence:** 5

**Summary:**

The paper first identifies a problem in previous works using synthetic images for bias mitigation—introducing a new source of bias due to the artifacts in synthesized images. To address this problem, the paper proposes From Fake to Real (FFR), a two-stage bias mitigation method. In the first stage, the model is pretrained on the balanced synthetic dataset. In the second stage, the model is finetuned on real data. Since FFR separates the synthetic and real data in two training stages, the identified problem is addressed. In the experiments, FFR outperforms existing methods in high-bias settings on three datasets.

**Strengths:**

1. The paper is well-motivated. The paper identifies the problem of previous works using synthetic datasets for debiasing—introducing a new bias source of synthetic artifacts, which is a very good motivation for the proposed method to address the problem.
2. The paper also presents a theoretical analysis (Theorem 1) of using synthetic datasets for debiasing in previous works.
3. The proposed method, two-stage training, is simple yet effective.
4. The paper is well-written and easy to follow.
5. The paper provides code for better transparency and reproducibility.

**Weaknesses:**

### Major Concerns

**Missing Comparison Methods**:
* The paper fails to add comparison methods to address the problem of the newly introduced synthetic artifact bias—using more bias groups in debiasing methods that use group labels: such as GroupDRO (Sagawa et al, 2020a), SUBG [1], DFR (Kirichenko et al, 2023), and Domain Independent (Wang et al., 2020c). While the authors argue that “in order to account for the new source of bias from $(B, G)$ where $G = \{ Real, Synthetic \}$, this approach doubles the number of bias groups ($| (B, G) | = | B || G | = 2 | B |$) which increases the optimization difficulty, reducing performance,” no experimental results are presented to support this claim. In fact, in a multi-shortcut (i.e., multi-bias) benchmark, Li et al. [2] show that the aforementioned four methods can successfully mitigate multiple biases simultaneously (see Table 5 in [2]).
* The paper also fails to compare with methods that infer group (or environment) labels, including LfF [3], George [4], EIIL [5], JTT [6], PGI [7], DebiAN [8], and CnC [9]. By inferring group labels instead of using the predefined one, these methods have the opportunity to mitigate the newly introduced artifact bias (or even other unknown biases).


**High bias settings are unrealistic**: I agree with the authors that the proposed FFR achieves better performance than existing approaches in the high-bias settings (e.g., 99.9% bias ratio). However, I think such extreme bias ratios (95% to 99.9% bias ratios in Figure 3) are too artificial to be found in real-world data. Thus, I encourage the authors to add experiments on real-world data without manipulating the bias ratios.

**Ablation study on the ordering of two stages**: The paper lacks a critical ablation study on the ordering of the two stages in FFR. Why not train on real data first and then train on the synthetic data? I think this alternative is more feasible—most computer vision models are pretrained on real-world data and further finetuning these models on balanced synthetic data is more practical.

**Biases in Stable Diffusion**    The paper uses Stable Diffusion (SD) v1.4 (“implementation details” on page 5) to generate the synthetic images. However, using SD to generate training has many problems:
* First, there is no guarantee that SD will faithfully follow the text prompts for image generation. In fact, SD and many other text-to-image models often struggle with compositional text prompts, e.g., spatial relationships [10] or attribute composition [11]. Thus, it is questionable whether SD can faithfully generate images by following the prompt template of “A photo of { bias } { class }” (“implementation details” on page 5).
* Even assuming that SD can faithfully follow the text prompt, new biases that are not controlled (e.g., explicitly specified) by the text prompts may also be introduced. For example, when generating female smiling images for CelebA HQ dataset, SD may generate stereotypical long-hair female images, leading to newly introduced hair length bias. Not mitigating the newly introduced biases has serious consequences—the amplification of the biases that are not mitigated in multi-bias scenarios [2].

### Minor Concerns
* The paper fails to discuss many works on “uncovering spurious correlations” (Section 2, Page 3). First, as mentioned above, many debiasing methods [3-9] that infer group labels are not discussed. Second, methods directly focused on bias detection are not discussed, such as methods based on clustering [12-14] or classification [15], generative models [16-18], explainability methods [19], and concepts [20,21].
* In the code, I found that the experiments use ImageNet pretrained weights to initialize the model. I wonder if this is consistent with the claim that the first stage of the FFR method “learns unbiased representations” for initialization.
* On the UTK-Face dataset, the authors “use age as the bias attribute and gender as the target attribute” (“Datasets” in Section 4, page 5). However, I don’t think this is reasonable because the age attribute is shown to be more difficult to learn than gender in two papers (see Figure 3 in [22] and Table 11 in (Ramaswamy et al., CVPR, 2021)). As shown in [3], the bias can only be learned if and only if the bias attribute is easier to learn than the target attribute.
* Regarding the synthetic datasets generated by SDv1.4, will they be released for reproducibility purposes?

### Minor Comments

* Section 4.3: Figure 5 reports our … -> Figure 4 reports our
* All quotation marks in the paper are misused (e.g., ”spurious” in Section 3, ”Big Dogs Indoors”, ”Small Dogs outdoors”, and ”toes” in Section 4.4). Please use ``’’ for quotation marks in LaTeX.
* The reference to “Kai Xiao, Logan Engstrom, Andrew Ilyas, and Aleksander Madry. Noise or signal: The role of image backgrounds in object recognition. ArXiv preprint arXiv:2006.09994, 2020.” (page 12) is outdated. The paper was accepted to ICLR 2021 [23].

### References

[1] Badr Youbi Idrissi, Martin Arjovsky, Mohammad Pezeshki, and David Lopez-Paz, “Simple data balancing achieves competitive worst-group-accuracy,” in CLeaR, 2022.

[2] Zhiheng Li, Ivan Evtimov, Albert Gordo, Caner Hazirbas, Tal Hassner, Cristian Canton Ferrer, Chenliang Xu, and Mark Ibrahim, “A Whac-A-Mole Dilemma: Shortcuts Come in Multiples Where Mitigating One Amplifies Others,” in CVPR, 2023.

[3] Junhyun Nam, Hyuntak Cha, Sungsoo Ahn, Jaeho Lee, and Jinwoo Shin, “Learning from Failure: Training Debiased Classiﬁer from Biased Classiﬁer,” in NeurIPS, 2020.

[4] Nimit S. Sohoni, Jared A. Dunnmon, Geoffrey Angus, Albert Gu, and Christopher Ré, “No Subclass Left Behind: Fine-Grained Robustness in Coarse-Grained Classification Problems,” in NeurIPS, 2020.

[5] Elliot Creager, Joern-Henrik Jacobsen, and Richard Zemel, “Environment Inference for Invariant Learning,” in ICML, 2021.

[6] Evan Z. Liu, Behzad Haghgoo, Annie S. Chen, Aditi Raghunathan, Pang Wei Koh, Shiori Sagawa, Percy Liang, and Chelsea Finn, “Just Train Twice: Improving Group Robustness without Training Group Information,” in ICML, 2021.

[7] Faruk Ahmed, Yoshua Bengio, Harm van Seijen, and Aaron Courville, “Systematic generalisation with group invariant predictions,” in ICLR, 2021.

[8] Zhiheng Li, Anthony Hoogs, and Chenliang Xu, “Discover and Mitigate Unknown Biases with Debiasing Alternate Networks,” in ECCV, 2022.

[9] Michael Zhang, Nimit S. Sohoni, Hongyang R. Zhang, Chelsea Finn, and Christopher Re, “Correct-N-Contrast: a Contrastive Approach for Improving Robustness to Spurious Correlations,” in ICML, 2022.

[10] Tejas Gokhale, Hamid Palangi, Besmira Nushi, Vibhav Vineet, Eric Horvitz, Ece Kamar, Chitta Baral, and Yezhou Yang, “Benchmarking Spatial Relationships in Text-to-Image Generation.” arXiv, 2022.

[11] “T2I-CompBench: A Comprehensive Benchmark for Open-world Compositional Text-to-image Generation,” NeurIPS Datasets and Benchmarks Track, 2023.

[12] Arvindkumar Krishnakumar, Viraj Prabhu, Sruthi Sudhakar, and Judy Hoffman, “UDIS: Unsupervised Discovery of Bias in Deep Visual Recognition Models,” in BMVC, 2021.

[13] Sabri Eyuboglu, Maya Varma, Khaled Kamal Saab, Jean-Benoit Delbrouck, Christopher Lee-Messer, Jared Dunnmon, James Zou, and Christopher Re, “Domino: Discovering Systematic Errors with Cross-Modal Embeddings,” in ICLR, 2022.

[14] Gregory Plumb, Nari Johnson, Angel Cabrera, and Ameet Talwalkar, “Towards a More Rigorous Science of Blindspot Discovery in Image Classification Models,” TMLR, 2023

[15] Saachi Jain, Hannah Lawrence, Ankur Moitra, and Aleksander Madry, “Distilling Model Failures as Directions in Latent Space,” in ICLR, 2023.

[16] Zhiheng Li and Chenliang Xu, “Discover the Unknown Biased Attribute of an Image Classifier,” in ICCV, 2021.

[17] Oran Lang, Yossi Gandelsman, Michal Yarom, Yoav Wald, Gal Elidan, Avinatan Hassidim, William T. Freeman, Phillip Isola, et al., “Explaining in Style: Training a GAN to explain a classifier in StyleSpace,” in ICCV, 2021.

[18] Jan Hendrik Metzen, Robin Hutmacher, N. Grace Hua, Valentyn Boreiko, and Dan Zhang, “Identification of Systematic Errors of Image Classifiers on Rare Subgroups,” in ICCV, 2023.

[19] Sahil Singla and Soheil Feizi, “Salient ImageNet: How to discover spurious features in Deep Learning?,” in ICLR, 2022.

[20] Shirley Wu, Mert Yuksekgonul, Linjun Zhang, and James Zou, “Discover and Cure: Concept-aware Mitigation of Spurious Correlation,” in ICML, 2023.

[21] Abubakar Abid, Mert Yuksekgonul, and James Zou, “Meaningfully Debugging Model Mistakes using Conceptual Counterfactual Explanations,” in ICML, 2022.

[22] Luca Scimeca, Seong Joon Oh, Sanghyuk Chun, Michael Poli, and Sangdoo Yun, “Which Shortcut Cues Will DNNs Choose? A Study from the Parameter-Space Perspective,” in ICLR, 2022.

[23] Kai Yuanqing Xiao, Logan Engstrom, Andrew Ilyas, and Aleksander Madry, “Noise or Signal: The Role of Image Backgrounds in Object Recognition,” in ICLR, 2021.

**Questions:**

I expect the authors to address my concerns listed in the weaknesses section:

* Adding more comparison methods.
* Adding experiments on real-world datasets with real spurious correlations, such as ImageNet background challenge [23] and ImageNet-W [2].
* Adding the ablation study on the ordering of two stages.
* Discussing the bias problem caused by SD.

---

> ### Author Response · Authors · 2023-11-17
>
> >The paper fails to add comparison methods to address the problem of the newly introduced synthetic artifact bias.
>
> We note that the reviewer is mistaken. GroupDRO, DFR, and resampling are applied on $(B, G)$ to address the synthetic-real bias. In fact, as we discuss in our work, this significantly improves the performance of synthetic augmentation methods (ASB and USB) due to precisely these methods addressing this issue. However, as we note in our results, they don’t effectively mitigate synthetic-real bias in high bias setting due to as the reviewer cited in our work “increase in the optimization difficulty”.
>
> >The paper also fails to compare with methods that infer group (or environment) labels, including LfF [3], George [4], EIIL [5], JTT [6], PGI [7], DebiAN [8], and CnC [9]. By inferring group labels instead of using the predefined one, these methods have the opportunity to mitigate the newly introduced artifact bias (or even other unknown biases).
>
> Note that the methods the reviewer suggests are developed to account for cases when bias labels are not available at training time, but we can always expect to have those labels in our setting since we can assume we know what images we generated. However, one could argue that the methods suggested by the reviewer are unsupervised methods, which tend to underperform those with supervision (i.e., those who assume they know what the bias groups are and aim to effectively take advantage of this information).  Indeed, one of our benchmarked methods is DFR which is state of the art in the unsupervised setting. While it doesn’t infer group labels on the training set, it uses a balanced labeled validation set to fine-tune a linear classifier.
>
> >These methods have the opportunity to mitigate the newly introduced artifact bias
>
> The real-synthetic group labels are trivially inferred because we know which samples are generated by a generator. While methods that infer group labels could be deployed to infer such labels, it would simply attempt to estimate labels that are already known. Nevertheless, even if we did not have these labels, identifying those with generator bias can be accomplished by simply deploying models that predict which images are generated by stable diffusion such as [1] which achieves a high accuracy.
>
> >High bias settings are unrealistic.
>
> We disagree with the reviewer. In fact, SpuCO animals naturally have a 95% bias and CelebA have many attributes with over 99% bias like “5-o-clock-shadow” and “Wearing_Lipstick”. However, we don’t test on these attributes as there are not enough samples in the test set for minority subgroups to estimate the worst group accuracy. Therefore, we pick Smiling, where there are enough samples in the validation and test set (at least 15 per group) and artificially vary the bias ratio.
>
> >First, there is no guarantee that SD will faithfully follow the text prompts for image generation.
>
> First, we note that our approach is generative model agnostic and that it will still work with future developments in generative models that address such issues.  Either way, it is clear that the synthetic data from Stable Diffusion on average generates correct images in our benchmarks since synthetic data augmentation, as we show in our results, significantly improves performance.  Moreover, SD has also shown great improvements in the quality of synthetic data generation. However, as we show in our work, it still leaves artifacts in images that could be exploited by recognition models, which is what our work primarily addresses.
>
> >New biases that are not controlled (e.g., explicitly specified) by the text prompts may also be introduced.
>
> We agree with the reviewer and discuss this issue in the limitations section. Indeed, future work could benefit from building fairer and more robust generative models. Either way, we note that our approach is generative model agnostic and that it will still work with future developments in generative models that address such issues.
>
> >The paper fails to discuss many works on “uncovering spurious correlations”.
>
> We thank the reviewer for pointing this out. We will expand our related section and discuss the relevance of the reviewer-cited work. Note, however, that none of these papers explore the same setting as our work, so are only loosely related.

---

> > ### Author Response · Authors · 2023-11-17
> >
> > >In the code, I found that the experiments use ImageNet pretrained weights to initialize the model. I wonder if this is consistent with the claim that the first stage of the FFR method “learns unbiased representations” for initialization.
> >
> > Note that we don’t claim that the model encodes unbiased representations after being initialized with imagenet weights. We make this claim about the model when it completes Stage 1: being trained on balanced synthetic data which then leads to unbiased representations.   In other words, the portion of the code you are referring to is for Stage 1 training, but not Stage 2 where the unbiased representations have been learned.
> >
> > >Ablation study on the ordering of two stages: The paper lacks a critical ablation study on the ordering of the two stages in FFR. Why not train on real data first and then train on the synthetic data? I think this alternative is more feasible—most computer vision models are pretrained on real-world data and further finetuning these models on balanced synthetic data is more practical.
> >
> >
> > Observe the results in the table below
> >
> > |                                            	| WA                 	| BA                 	|
> > |------------------------------------------------|------------------------|------------------------|
> > | Stage 1                                  	| 54.3               	| 78.7               |
> > | Stage 2                                  	| 3.7                  	| 57.6               |
> > | Stage 2 --> Stage 1               	| 51.7               	| 76.5               |
> > | Stage 1 --> Stage 2: FFR (ours)       | 74.9 3.1           	|  84.5              |
> >
> > Note that switching the order doesn’t perform as well as our ordering (FFR).
> >
> > >On the UTK-Face dataset, the authors “use age as the bias attribute and gender as the target attribute” (“Datasets” in Section 4, page 5). However, I don’t think this is reasonable because the age attribute is shown to be more difficult to learn than gender in two papers (see Figure 3 in [22] and Table 11 in (Ramaswamy et al., CVPR, 2021)). As shown in [3], the bias can only be learned if and only if the bias attribute is easier to learn than the target attribute.
> >
> > We agree with the reviewer's insight that when the bias attribute is more difficult to learn, it is less likely for the model to be biased.  However, in our experiment, we use a split in UTK-Face where age is split into two categories (young versus old). The differences between these two groups are easily discernable that performance degrades on gender prediction when the bias increases, as indicated in Figure 2 in our paper. The performance won’t degrade unless the model has learned the bias as studied in the paper referenced by the reviewer ([3]). Moreover, the evidence that the reviewer cited from (Ramaswamy et al., CVPR, 2021) is solely conducted on CelebA which has a limited distribution of age differences.
> >
> > >Regarding the synthetic datasets generated by SDv1.4, will they be released for reproducibility purposes?
> >
> > Yes.
> >
> > >Adding experiments on real-world datasets with real spurious correlations, such as ImageNet background challenge [23] and ImageNet-W [2].
> >
> > The datasets used in our work are standard bias mitigation datasets and they are real datasets with real spurious correlations. However, both datasets that the reviewer cited in their review use synthetic biases, so we fail to see how their inclusion will add any further insights.
> >
> > [1] Riccardo Corvi, Davide Cozzolino, Giada Zingarini, Giovanni Poggi, Koki Nagano, and Luisa Verdoliva. On the detection of synthetic images generated by diffusion models, ICASSP 2023

---

### Official Review · Reviewer_DngP · 2023-10-31

**Soundness:** 2 fair
**Presentation:** 3 good
**Contribution:** 2 fair
**Rating:** 5
**Confidence:** 4

**Summary:**

This work considers the problem of bias mitigation on image classification tasks and tackles it using balanced synthetic data. More specifically, authors claim naively combining real data with balanced synthetic might not be an efficient way to deal with bias due spurious correlations because the model might exploit potentially existing differences between the real and synthetic distributions when making predictions. To address this issue, authors then propose From Fake to Real (FFR), a two step training framework for bias mitigation. In FFR, a model is trained with balanced synthetic data only, followed by training with samples from the original real distribution. Authors used an off-the-shelf diffusion model to generate balanced synthetic data and compared FFR with other approaches in the literature that considered training a model in a single step by combining synthetic and real data. Additionally, authors combined the proposed training framework (as well as the baselines) with techniques to mitigate bias based such as GroupDRO. The empirical evaluation considered a single architecture (ResNet-50) and three datasets. Overall, results showed that FFR is more effective at bias mitigation as per two metrics, worst group accuracy and balanced accuracy.

**Strengths:**

- The manuscript is well-written
- The work is well-motivated and the authors contextualized their contributions with respect to prior work
- Contributions are potentially relevant as the work touches on a topic of utmost importance for the machine learning community
- The proposed technique is intuitive and seems relatively easy to implement/reproduce in case one assumes access to the same generative model used in the manuscript
- The empirical evaluation takes into account multiple baselines and datasets

**Weaknesses:**

- FFR, the central contribution of this work, is based on the assumption that a model can differentiate between real and synthetic images and is likely to leverage correlations that are not stable between these two sources of data to make predictions, which could lead to biased predictions in the real dataset. However, there is no discussion or result (theoretical or empirical) in the manuscript to support this rather strong claim. (See the next section for questions related to this point)

- The authors consider a two-step training approach, but only report and discuss in the manuscript the performance of the second step, making it hard to fully understand whether (and to what extent) the model is successful at prediction after being trained with balanced synthetic data only. Moreover, the manuscript does not report/investigate whether models are encoding biases or not after training with balanced synthetic data.

- This contribution heavily relies on the use of synthetic datasets, however, there is no discussion regarding the quality of the generated data.  (See the next section for questions related to this point)

- No error bars or confidence intervals were reported in the results and it seems that a single training was performed for each model, making to hard to evaluate how robust the finding from the experimental evaluation are.

- It is not clear from the manuscript whether all approaches trained with synthetic data had access to the same amount of synthetic instances at training time. If not, the observed conclusions from the experiment could have been confounded by the different number samples in each training dataset.

- The authors used an off-the-shelf generative model to try to approximate the real data distribution, but it seems to me that training specialized models for each dataset could yield higher quality data while also helping to mitigate bias due to unbalanced subgroups [1].

[1] Ktena et al., Generative models improve fairness of medical classifiers under distribution shifts, 2023.

**Questions:**

- What if the considered model class does not capture the artifacts that make real and synthetic distinguishable? How would the improvements obtained with FFR reported in the manuscript would be impacted in this case?

- How would the performance of the approach be impacted by low-quality generated data? More specifically, what would happen in case the generative is not able to generate some classes in the original dataset with good quality?

- Why were no augmentations used to train the ResNet-50s? Could this cripple the ERM+No synthetic data baseline, thus rendering the comparison unfair?

- The authors mentioned in the limitation section that the employed datasets are quite small. Given that and the fact that no augmentations were used, could the improvements observed on top of the baseline that does not have access to synthetic data be simply due to the fact the training data is larger after synthetic data is added?

---

> ### Author Response · Authors · 2023-11-17
>
> >FFR, the central contribution of this work, is based on the assumption that a model can differentiate between real and synthetic images and is likely to leverage correlations that are not stable between these two sources of data to make predictions, which could lead to biased predictions in the real dataset. However, there is no discussion or result (theoretical or empirical) in the manuscript to support this rather strong claim. (See the next section for questions related to this point)
>
> We conduct a theoretical analysis where we prove that assuming these artifacts exist, then models should pick up on these correlations as the dataset distribution that combines real and synthetic data would be biased. Refer to Theorem 1 for more details. In addition, we cite prior work [1] that shows how artifacts are present in even modern generative models like Stable Diffusion. Therefore, our method, by separating the two data sources, achieves significant improvement in performance. We provide further insight into this phenomenon by providing the worst group accuracy over the synthetic data in addition to real data for our benchmarked synthetic augmentation methods (USB and ASB) and our method (FFR). Indeed, if the model leveraged such artifact-based correlations, then the model will be biased not only on the real data but also on the synthetic data. Observe the results in the table below:
>
> |              | Real Data |      | Synthetic Data |      |
> |--------------|:---------:|------|:--------------:|------|
> |              | WA        | BA   | WA             | BA   |
> | USB        | 54.7      | 78.9 | 89.3           | 83.1 |
> | ASB        | 39.9      | 73.4 | 68.7           | 85.2 |
> | FFR (ours) | 74.9      | 84.5 | 94.5           | 97.8 |
>
> Note that our method not only achieves the best worst group accuracy on real data but also on synthetic data. This means that our method is not relearning the bias on both real and synthetic data but is effectively using the synthetic data to mitigate the bias and generalize to real data.
>
> >How would the performance of the approach be impacted by low-quality generated data? More specifically, what would happen in case the generative is not able to generate some classes in the original dataset with good quality?
>
>
> We perform the pixelate effect from [2]. We provide severity 5 which is the highest level. Observe the results of the synthetic augmentation methods below:
>
> |  | Severity  | 0 | Severity | 5 |
> |---|---|---|---|---|
> |  | WA | BC | WA | BC |
> | USB | 60.4 | 82.0 | 49.3 | 78.6 |
> | ASB | 62.8 | 82.3 | 55.2 | 78.3 |
> | FFR | 76.1 | 86.0 | 70.9 | 84.6 |
>
> Note that while the performance drops due to the deterioration of quality, our method remains the most effective among the other synthetic augmentation methods.
> >The authors consider a two-step training approach but only report and discuss in the manuscript the performance of the second step.
>
> Observe the results of step 1 and 2 broken down on SpuCO animals:
>
> |                                            	| WA                 	| BA                 	|
> |------------------------------------------------|------------------------|------------------------|
> | Stage 1                                  	| 54.3               	| 78.7               |
> | Stage 2                                  	| 3.7                  	| 57.6               |
> | Stage 1 --> Stage 2: FFR (ours)       | 74.9            	|  84.5              |
>
>
> Note that Stage 2 (second line in table) by itself yields poor performance. This is expected as Stage 2 amounts to training a model with ERM on the biased data without pertaining on synthetic data. Thus, the model, as expected, learns the bias. Training with Stage 1 by itself (first line in the table) while improving performance over Stage 2 by itself, doesn't match the performance of FFR (Stage 1 → Stage 2). This is likely due to the real and synthetic data distribution gap. Therefore, following up Stage 2 with Stage 1 is important to bridge the performance gap.
>
>
> [1] Riccardo Corvi, Davide Cozzolino, Giada Zingarini, Giovanni Poggi, Koki Nagano, and Luisa Verdoliva. On the detection of synthetic images generated by diffusion models, ICASSP 2023
>
> [2] Michaelis, Claudio and Mitzkus, Benjamin and Geirhos, Robert and Rusak, Evgenia and  Bringmann, Oliver and Ecker, Alexander S. and Bethge, Matthias and Brendel, Wieland, Benchmarking robustness in object detection: Autonomous driving when winter is coming

---

> > ### Author Response · Authors · 2023-11-17
> >
> > >No error bars or confidence intervals were reported in the results.
> >
> > In our plots we report confidence intervals, where we obtain clearly better performance than prior work. These intervals are computed over 3 runs We don’t report these intervals in the table due to space constraints, but can include them in a revision.
> >
> > >It is not clear from the manuscript whether all approaches trained with synthetic data had access to the same amount of synthetic instances.
> >
> > All approaches had the same number of synthetic data used for training. We will clarify this in the final version of the paper.
> >
> > >The authors used an off-the-shelf generative model to try to approximate the real data distribution, but it seems to me that training specialized models for each dataset could yield higher quality data while also helping to mitigate bias due to unbalanced subgroups
> >
> > Note that our work focuses on high-bias settings where training specialized models is infeasible, given that there are very few samples available for certain subgroups. Therefore training generative models from scratch or fine-tuning pretrained ones on the minority subgroups will not yield higher-quality synthetic images. Either way, the issue of artifacts will persist, which is what our work is primarily addressing.
> >
> > >Why were no augmentations used to train the ResNet-50s? Could this cripple the ERM+No synthetic data baseline, thus rendering the comparison unfair?
> >
> > We don’t conduct augmentation to isolate the effect of spurious correlations on performance. However, we conduct a comparison between ERM + FFR and ERM + no-synthetic data on CelebA when augmentation is used,
> > |                                        | CelebA |      |
> > |----------------------------------------|:------:|------|
> > |                                        | WA     | BA   |
> > | ERM + No Synthetic Data                | 37.6   | 57.3 |
> > | ERM + No Synthetic Data + Augmentation | 34.2   | 68.8 |
> > | ERM + FFR                              | 76.6   | 86.6 |
> > | ERM + FFR + Augmentation               | 77.6   | 87.3 |
> >
> > Note how augmentation slightly amplifies the bias of ERM (lower worst group accuracy) while slightly improving the performance of our method.
> >
> >
> > [1] Riccardo Corvi, Davide Cozzolino, Giada Zingarini, Giovanni Poggi, Koki Nagano, and Luisa Verdoliva. On the detection of synthetic images generated by diffusion models, ICASSP 2023
> >
> > [2] Michaelis, Claudio and Mitzkus, Benjamin and Geirhos, Robert and Rusak, Evgenia and  Bringmann, Oliver and Ecker, Alexander S. and Bethge, Matthias and Brendel, Wieland, Benchmarking robustness in object detection: Autonomous driving when winter is coming

---

### Official Review · Reviewer_RuLj · 2023-10-31

**Soundness:** 2 fair
**Presentation:** 2 fair
**Contribution:** 2 fair
**Rating:** 6
**Confidence:** 3

**Summary:**

The paper presents a method to tackle spurious correlations present in training data, that translate in subpar performance when evaluation tasks do not show these spurious correlations. To do so, the paper proposes using a large-scaled pretrained generative image model to sample for the inbalanced classes, which allows to produce a balanced dataset that does not show the spurious correlation. Despite this, the paper argues (and shows through experiments), that the approach of either training with only generative samples or with the naive combination of synthetic and real is suboptimal, as the model can exploit the shortcut solution of distinguish between real and fake data to gain information about the training task. To overcome this, the paper proposes a two stage method of 1) first training with synthetic data and then 2) finetuning with real data, which results in improved performance when compared to alternatives.

**Strengths:**

- The motivation is sound and is supported by simple but complete theory.
- The method is simple and relatively easy to apply, although it requires large-scale generative image models, sampling from them and a two stage training process.
- Quantitative results show good performance when compared to simple alternatives, showing that when using synthetic data to rebalance the dataset, the two stage method improves performance significantly.
- Qualitative results (Figure 5) illustrate that the proposed method is more able to ignore spurious correlations than alternatives, as the saliency maps are concentrated in the classification task at hand.

**Weaknesses:**

- The main issue of the paper is that it ignores the (large-scale) datasources used to train the generative model, and how they can be leveraged as simple baselines for the proposed method, by either finetuning a pretrained classification model (like OpenCLIP [1]) or retrieving images from these datasets to reduce the spurious correlations. Alternatively, a fair comparison would be to use a generative model trained only on the datasets being evaluated.

- In this regard, the model does not compare against using a pretrained backbone model on the same or similar large-scale datasources (LAION) and then testing it zero-shot, finetuning it, or learning a shallow classifier on top. This is a simple baseline with similar trade-offs as using a large scale generative model in terms of off-the-shelf availability, but has the advantage of not requiring sampling nor a two stage method. It may be the case that large-scale classification models are already able to disentangle and ignore the spurious correlations. Does a pretrained model on LAION-5B for classification, and then lightly finetuned for these tasks perform better or worse than the proposed full method?

- As pointed out in the limitations, the experiments are performed on a "toyish" set of tasks with small amounts of data and simple biases, while the method uses large-scale data. These simple biases showcased may not be as prevalent in the training dataset used for the generative model (LAION-5B), as LAION will likely have much greater variation. For example, it is likely that the smiling bias in people is more prevalent in CelebA HQ than in LAION, and a model trained on CelebA HQ is likely to pick it up because CelebA doesn't have much variation. Although how to best retrieve and use this data to tackle bias is out of the scope of the paper, it would be good to have simple baselines, like retrieving all or a subset of {class} images from LAION and training the first stage with that.

- The experiment does not show results for FFR without rebalancing, which is a simpler setting, as one doesn't need to know the {bias} attributes that cause the spurious correlations to generate images. It has been shown that just randomization can tackle domain missmatch and alleviate spurious correlations [2], making models perform well during evaluation, and this effect is not considered. Given that the generative model is powerful, it is possible that it is introducing extra variation (e.g. lots of different poses, background...), that biases the model to ignore the spurious correlation. To test that the hypothesis that *rebalancing* is the main driver of improved performance and not *extra random diversity* gained by the usage of large scale generative models, it would be good to compare the proposed rebalanced sampling method for the prompts to random sampling (i.e. instead of A photo of {bias} {class}, A photo of {class}, or A photo of {random word} {class}).

Despite these concerns, which if addressed, would make for a stronger paper, I believe the paper as it stands adds value and will be of interest to the community.


[1] https://github.com/mlfoundations/open_clip
[2] https://arxiv.org/abs/1710.06425

**Questions:**

See weaknesses.

Typo: Uncovering Spurious Correlations. In our work, we are interested *in* mitigating

---

> ### Author Response · Authors · 2023-11-17
>
> >The main issue of the paper is that it ignores the (large-scale) datasources used to train the generative model, and how they can be leveraged as simple baselines for the proposed method, by either finetuning a pretrained classification model (like OpenCLIP [1]) or retrieving images from these datasets to reduce the spurious correlations. Alternatively, a fair comparison would be to use a generative model trained only on the datasets being evaluated.
>
> We thank the reviewer for the comment.  First, we note that many large scale generative models are trained on non-public datasets, which means that there are already many problems for which we may be able to take advantage of the pretrained model, but not the data.  As these types of models continue to be explored, we expect that this will only increase during time.
>
> Second, we note that retrieving images from these datasets to reduce the spurious correlations is not a trivial task. It is computationally expensive to sort and categorize through datasets like LAION given their sheer size, and rare categories may be challenging to even detect as inaccurate models used to filter them may result in many false positives. However, using generative models like Stable Diffusion is much cheaper, and as our work demonstrates, can be done in an automated way and still boost performance. Therefore, we deem methods that rely on filtering these datasets as out of the scope of our work.
>
> > or by finetuning a pretrained classification model (like OpenCLIP [1])
>
> We agree with the reviewer that this is an interesting question. In fact, recent work [1] has noted that fine-tuning models like CLIP has to be done with care so the bias is not learned. Thus, we would expect that our work would also benefit CLIP, as it could simply be used as a drop-in replacement.  However, we explore ResNet-50 models for a fair comparison to prior work [2,3,4].
>
> > Alternatively, a fair comparison would be to use a generative model trained only on the datasets being evaluated
>
> One of our main paper observations is that synthetic data is most useful in high bias settings (99.9% bias) where minority subgroups have as few samples as few as 10 samples. Thus, obtaining useful synthetic data for the minority subgroups by training generative models only on the evaluated dataset is likely impossible. Thus, this motivates our use of Large Scale generative models, where these samples may have been seen and learned due to the significant training datasets.   Additionally, we note that all the methods we compare to use the synthetic data during training, so our comparisons are fair.
>
> Overall, we note that our main contribution is noting a critical issue using generative models for bias mitigation. Our analysis and method are agnostic to the generative model being used as artifacts are a consistent problem across all current generative models [5]
>
> >The experiment does not show results for FFR without rebalancing, which is a simpler setting, as one doesn't need to know the {bias} attributes that cause the spurious correlations to generate images. It has been shown that just randomization can tackle domain missmatch and alleviate spurious correlations [2], making models perform well during evaluation, and this effect is not considered. Given that the generative model is powerful, it is possible that it is introducing extra variation (e.g. lots of different poses, background...), that biases the model to ignore the spurious correlation. To test that the hypothesis that rebalancing is the main driver of improved performance and not extra random diversity gained by the usage of large scale generative models, it would be good to compare the proposed rebalanced sampling method for the prompts to random sampling (i.e. instead of A photo of {bias} {class}, A photo of {class}, or A photo of {random word} {class}).
>
> We thank the reviewer for this comment. Indeed, this is an interesting question. We will include such questions in the revised version of our work.
>
> [1] Debiased Fine-Tuning for Vision-Language Models by Prompt Regularization, Beier Zhu1, Yulei Niu2*, Saeil Lee3, Minhoe Hur4, Hanwang Zhang, in AAAI 2023
>
> [2] Kirichenko, Polina, Pavel Izmailov, and Andrew Gordon Wilson. "Last layer re-training is sufficient for robustness to spurious correlations."  ICLR 2023
>
> [3] Sagawa, Shiori, et al. "Distributionally robust neural networks for group shifts: On the importance of regularization for worst-case generalization." arXiv preprint arXiv:1911.08731 (2019).
>
> [4] Badr Youbi Idrissi, Martin Arjovsky, Mohammad Pezeshki, and David Lopez-Paz, “Simple data balancing achieves competitive worst-group-accuracy,” in CLeaR, 2022.
>
> [5] Riccardo Corvi, Davide Cozzolino, Giada Zingarini, Giovanni Poggi, Koki Nagano, and Luisa Verdoliva. On the detection of synthetic images generated by diffusion models, ICASSP 2023